# Optimal Stopping Methods for Investment Decisions: A Literature Review

**Zhenya Liu** [1,2,3,*] **and Yuhao Mu** [1]

1   School of Finance, Renmin University of China, Beijing 100872, China
2   China Financial Policy Research Center, Renmin University of China, Beijing 100872, China
3   CERGAM, Aix-Marseille University, CEDEX 07, 13284 Aix-en-Provence, France
*   Correspondence: zhenya.liu@ruc.edu.cn

**Abstract:** Investors decide the best time to take a given action by maximizing their utility function while taking into account current information and the underlying process in the optimal stopping model. Option pricing, sequential analysis, disorder problems, and other problems requiring time decision-making are all examples of this type of problem. A lot of literature has studied optimal stopping models and put forward the corresponding solutions. Investors in financial markets must also know when to buy and sell, so timing is crucial. This paper presents a classified review of the literature on optimal stopping models, followed by a summary of the strategies that can be used in financial markets to make investment decisions using optimal stopping methods.

**Keywords:** optimal stopping; sequential analysis; disorder problem; regime-switching

## 1. Introduction

The financial market is fraught with uncertainty, and a lot of literature studies how to identify and control risks when uncertainty exists, for example, Kou et al. (2014) and Locurcio et al. (2021). Uncertainty causes investors to hesitatingly make investment decisions, or even to make erroneous ones, such as the well-known disposal effect. Here we introduce the optimal stopping method to help investors make rational decisions when facing uncertainty.

The optimal stopping method is important in investment. This method uses the stochastic model to describe the stock market's trend and set up an analytical framework for observing and understanding the logic behind stock market changes in a scientific way. After using real market data to describe how the market changes in a stochastic model, investors would predict the market movement in the future and make relative investment decisions based on the model. Such methods include Merton's model (Merton 1969) and the optimal stopping models to be discussed in this article. The optimal stopping models suggest when investors should buy and sell, and Merton's model solves the problem of how much investors should buy and sell.

As a special dynamic programming model, the optimal stopping model consists of an objective function and constraints. The objective function can be the utility or return function of investors, and the constraints are the underlying stochastic process. Take an infinite American put option with strike price K as an example. Suppose that the stock price X obeys geometric Brownian motion: $dX_t = rX_t dt + \sigma X_t dB_t, X_0 = x$, which is a constraint. Investors hold the option and look for the optimal time $\tau$ to execute to maximize the discounted yield of maturity (objective function):

$$V(x) = \sup_{\tau \geq 0} \mathbf{E}\left[e^{-r\tau}(K - X_\tau)^+\right] \quad (1)$$

The solution of the problem is: $\tau^* = min\left\{\tau : X_\tau \leq \frac{2rK}{2r+\sigma^2}\right\}$. In other words, it is the best choice to exercise the option immediately only when the price falls below a critical

value. The set where the price is higher than the critical value is called the continuation set, and the area where the price is lower than the critical value is called the stopping set. In an optimal stopping model, we often obtain a boundary that divides the space into two sets.

The remainder of this paper is organized as follows: Section 2 briefly summarizes the development and application of optimal stopping models. Section 3 presents the classification of optimal stopping models, which is classified by the constraint conditions (underlying processes) and objective functions. Section 4 includes the analytical and numerical solutions of optimal stopping models used in investment decision making, which are divided into five strategies: Sequential analysis, disorder problem, optimal prediction, buy-low and sell-high, and regime-switching. Section 5 introduces the market performance of some strategies. Finally, Section 6 concludes.

## 2. Literature Review

The optimal stopping model originated from Wald (1947) and Wald and Wolfowitz (1948) and was used by Wald for sequential analysis, that is, the sampling was not stopped until the results were significant enough. Later research focused on the method of solving the optimal stopping model. A martingale method for studying discrete-time optimal stopping models is proposed by Snell (1952). On the other hand, applying a dynamic programming framework to solve the optimal stopping models is also useful, see Arrow et al. (1949). Dvoretzky et al. (1953) earlier studied optimal stopping models under continuous time, and solved the problem of sequential testing, that is, to decide whether the actual data are more likely to follow which of the two stochastic processes as soon as possible.

A breakthrough of optimal stopping models is to combine them with free boundary problems. Now, most optimal stopping models in continuous time are solved by this method. McKean (1965) and Mikhalevich (1958) introduced the smooth fit condition into optimal stopping models, which began to be associated with the free boundary problems. The boundary of an optimal stopping model divides the parameter space into two regions: continue to wait the best time and stop immediately. The boundary and the solution of a free boundary problem are determined at the same time. In this strand of literature, authors often pre-specify the shape of the boundary in a general form and then solve and verify it. Shiryaev (1961) and Shiryaev (1963) began to study a disorder problem, that is, to quickly identify the time when the parameters of the stochastic process changed, and solve it through the corresponding free boundary problem. Beibel and Lerche (1997) derived the optimal strategy of selling stocks by maximizing the stock price after discount under the framework of the disorder problem.

Most of the initial optimal stopping models are one-dimensional, that is, infinite optimal stopping models. In practice, decisions often need to be made within a period. To address this issue, some subsequent studies naively added the time dimension into the model. However, it would increase the difficulty of solving a two-dimensional problem, because the free boundary problem needs to be solved by a partial differential equation. It is challenging to obtain an analytical solution for this problem. McKean (1965) firstly analyzed the pricing of the American option with finite horizon, and transformed it into a free boundary problem. Myneni (1992) summarized two methods of using the free boundary problem and variational inequality to price the American option with finite horizon, and put forward the problem of whether the solution is unique or not. Peskir (2005a) proved the uniqueness of the solution and gave an analytical solution of the price of the American option with the finite horizon. This proposed method also can be extended to other finite horizon problems. In addition to adding the dimension of time, several studies investigate the impact of the maximum historical price on decision-making. Other studies consider adding the maximum process to the utility function of the optimal stopping model, Shepp and Shiryaev (1993) introduced the Russian option, the payment of which is the highest price of the stock during the option holding period, and obtained an analytical solution through the free boundary problem and reasonable speculation on the form of the solution. Peskir (1998) presented the maximum principle, which is used to solve a class of

optimal stopping models with the maximum process. Egami and Oryu (2017) provided a canonical process to solve a more extensive class of optimal stopping models involving maximum process through dimension reduction.

The most commonly used stochastic process for continuous-time optimal stopping models is geometric Brownian motion or one-dimensional diffusion process. Geometric Brownian motion has limitations in characterizing stock price movement, and the parameters of it do not vary with time. To better characterize stock price movement, complex stochastic processes are considered, such as the Lévy process, stochastic volatility model, etc. A regime-switching model where price is modeled by adjoining a hidden Markov process to the classical geometric Brownian motion is a proper alternative. In such a flexible model, the parameters in each state are different, and the transformation time between different states is random. For example, the underlying process in the disorder problem mentioned above is a kind of regime-switching models. This model can well describe the transformation between bull and bear markets in the real world. Zhang (2001) and Guo and Zhang (2005) studied the investment decision-making problem in a regime-switching framework. Guo (2001), Buffington and Elliott (2002) and Boyle and Draviam (2007) studied option pricing problems in a regime-switching framework.

The optimal stopping model is widely used in many research fields. For example, it is used to decide when to stop clinical trials in medical science (Jennison and Turnbull 1999); in engineering, it provides optimal policy for maintenance operation (Machida and Miyoshi 2017); as for bioeconomics, the optimal stopping model is applied to find the optimal harvest policy (Sarkar 2009); in finance, the optimal stopping model is widely used for option pricing and asset-selling/buying problems. This paper focuses on the optimal stopping methods for investment decisions for two reasons, one is that most people are more familiar with financial markets, and timing is very important for investment decisions, the other is that the financial market data are sufficient to estimate the model parameters.

This paper mainly focuses on the optimal stopping models of continuous stochastic processes. From the practical perspective, this paper aims not only to provide a series of investment strategies based on optimal stopping, but also investigate the solutions of various stopping problems related to option pricing and derivatives hedging. Peskir and Shiryaev (2006) presented a very detailed summary of the solution and classification of optimal stopping models; however, this book did not go through the relevant applications of investment decision-making. Inspired by the summary of the basic knowledge of optimal stopping problems in Peskir and Shiryaev (2006), some new development and application of optimal stopping problems are summarized as follows.

## 3. Classification of Optimal Stopping Models

### 3.1. Constraint Conditions (Underlying Process)

Some commonly used stochastic processes in the optimal stopping models are present in Table 1. Process 1 is the standard Brownian motion, which is the simplest case. Most problems are analyzed from the standard Brownian motion or by transforming the stochastic process into the standard Brownian motion through the change of space or change of time. Standard Brownian motion serves as a benchmark for analyzing sophisticated random systems. This is the simplest random process, which can be used to explore complex optimal stopping models and offer a benchmark for expansion. However, the real world is far more complex, and the standard Brownian motion rarely describes it. Process 2 is the geometric Brownian motion, the most commonly used in the literature. Geometric Brownian motion is nonnegative, thus it is widely used to model asset prices and it is easy to analyze and somewhat realistic. However, geometric Brownian motion's parameters are time-invariant, thus subsequent stochastic processes add time-varying parameters. Process 3 is a one-dimensional diffusion process, and almost all the stochastic processes in the table are of this type. Process 3 generally appears in the general solution of a class of optimal stopping models. However, as Process 3 represents a vast class of stochastic processes, its general solution is hard to find, especially for complex issues. Process 4 is usually applied

for sequential analysis, that is, the drift term of the stochastic process is unknown, and the actual data is used to test and identify the drift term from two alternative parameters.

**Table 1.** Stochastic processes used in optimal stopping models.

| Label | Underlying Process | Note | References (Part) |
|---|---|---|---|
| 1 | $X_t = B_t$ | $B_t$: Standard Brownian motion, same as below | Graversen et al. (2001) |
| 2 | $dX_t = \mu X_t dt + \sigma X_t dB_t$ | $X_0 = x$, same as below | Shiryaev et al. (2008) |
| 3 | $dX_t = \mu(X_t)dt + \sigma(X_t)dB_t$ | Most are of this type or variations of this type | Dayanik and Karatzas (2003), Egami and Oryu (2017) |
| 4 | $dX_t = a_t X_t dt + \sigma X_t dB_t$ | $a_t \in \{a_h, a_l\}$, $\mathbf{P}(a_0 = a_h) = \pi_0$ $\mathbf{P}(a_0 = a_l) = 1 - \pi_0$ | Van Khanh (2012) |
| 5 | $dX_t = \mu(t)X_t dt + \sigma X_t dB_t$ | The drift changes from $\mu_1$ to $\mu_2$ at $\theta$ $\mathbf{P}(\theta > t) = (1 - p)e^{-\lambda t}$ | Beibel and Lerche (1997), Karatzas (2003) |
| 6 | $dX_t = X_t \mu_{\epsilon(t)} dt + X_t \sigma_{\epsilon(t)} dB_t$ | $(\mu_{\epsilon(t)}, \sigma_{\epsilon(t)}) = \begin{cases} (\mu_1, \sigma_1), & \epsilon(t) = 1 \\ (\mu_2, \sigma_2), & \epsilon(t) = 2 \end{cases}$ $\tau_i$ is the time leaving state $i$: $\mathbf{P}(\tau_i > t) = e^{-\lambda_i t}, \quad i = 1, 2$ | Zhang (2001), Guo (2001) |
| 7 | $dX_t = (r - \delta(S_t, Y_t))X_t\,dt + \sigma(S_t, Y_t)X_t\,dB_t$ | $S_t = s \vee \max_{0 \le u \le t} X_u$ $Y_t = y \vee \max_{0 \le u \le t}(S_u - X_u)$ | Gapeev and Rodosthenous (2014, 2016) |
| 8 | $dX_t = A_t \mu dt + dB_t$ | $A_t$ can be 1 or $-1$ by human intervention | Dalang and Vinckenbosch (2014) |
| 9 | $dX_t = \mu(X_t)dt + B_t$ | $\mu(x) = \begin{cases} \mu_1, & \text{for } x < 0 \\ \mu_2, & \text{for } x \ge 0 \end{cases}$ | Mordecki and Salminen (2019a) |
| 10 | $dX_t = \sigma(X_t)dB_t$ | $\sigma(x) = \begin{cases} \sigma_1, & x < 0 \\ \sigma_2, & x \ge 0 \end{cases}$ | Mordecki and Salminen (2019b) |

Process 5 is a special case of regime-switching and is used for disorder problems. Process 4 and Process 5 seem to be similar to one another. The specific forms of the stochastic processes are unknown and need to be inferred from the realized data. While the parameters of Process 4 are time-invariant, those of Process 5 are time-varying. Process 6 is a two-dimensional regime-switching model. In Section 3, the definition of N-dimension will be given, that is, the parameters switch in different states. Process 6 is more realistic and involved. It shows, for instance, how the stock market's bull and bear cycles occur in alternating fashion. Unfortunately, this process is usually difficult to deal with. Process 7 is a stochastic process used to construct a new option. The drift term and diffusion term are related to the maximum drawdown and maximum process. This design makes the stochastic process path dependent, which may be associated with behavioral finance, such as anchoring effect. When it comes to Process 8, timing is decided to intervene in a stochastic process, and parameters are artificially selected to maximize the objective function. The parameters of the previous processes cannot be controlled. Process 6 describes a case where

the stochastic process can be controlled. As for Processes 9 and 10, the reference gives the analytical solution to the corresponding optimal stopping model but the corresponding financial application is not given.

### 3.2. Problems and the Corresponding Objective Functions

The objective function of the optimal stopping model is closely related to the problem to be studied and the choice of the utility function is essential in formulating the problem. A good objective function can not only describe the problem appropriately but also simplify the solution steps. The optimal stopping models are designed for problems which can be classified into six categories: the classical sequential analysis problem, disorder problem, optimal prediction problem, buy-low and sell-high problem, option pricing problem, and others.

#### 3.2.1. Sequential Analysis

Assume that the actual data are generated by a stochastic process, the parameters of which are binary random variables, and the prior distribution of parameters is known. As time goes on, the number of observations increases and the posterior distribution of parameters keeps updating. The sequential analysis problem is to quickly as well as accurately judge the parameters from the alternatives. Take Process 4 as an example (for details, see Peskir and Shiryaev (2006)):

$$V(\pi_0) = \inf_{(\tau,d)} \mathbf{E}_{\pi}\left[(\tau + a\mathbf{1}_{\{d=0,a_t=a_h\}} + b\mathbf{1}_{\{d=1,a_t=a_l\}})\right] \tag{2}$$

where $\mathbf{1}$ is an indicator function, which is 1 if the condition is satisfied, otherwise 0. $d = 0$ when $a_t = a_l$ is accepted, whereas similar. $a$ and $b$ represent the penalty for each type of error, and $\tau$ indicates that the quicker the stochastic process is identified, the better. In this problem, we generally define a new stochastic process $\pi_t$ to depict a posterior probability process, and the problem can be transformed into a problem about $\pi_t$.

#### 3.2.2. Disorder Problem

This problem is analyzed under the setting of Process 5. Take the financial market as an example. Assume that it is a bull market at present, and the time when the bull market turns into a bear market is a random variable. The problem for investors is to quickly identify the moment of bull and bear markets transitions and make timely decisions. Here are two objective functions that measure the quickest detection:

Shiryaev (2002)

$$V(p) = \inf_{\tau}\left\{\mathbf{P}(\tau < \theta) + c\mathbf{E}[(\tau - \theta)^+]\right\} \tag{3}$$

Karatzas (2003)

$$V(p) = \min_{\tau} \mathbf{E}|\theta - \tau| \tag{4}$$

where $\mathbf{E}|\theta - \tau| = \mathbf{E}[(\tau - \theta)^+ + (\theta - \tau)^+] = \mathbf{E}[(\tau - \theta)^+] + \mathbf{E}[\theta] - \mathbf{E}[\tau \wedge \theta]$

From the perspective of investors, they aim to maximize their utility function instead of identifying the market transition time as soon as possible. The utility function is shown as:

$$V(x) = \sup_{\tau \geq 0} \mathbf{E}[U(X_\tau)] \tag{5}$$

Gapeev and Peskir (2006) gave the solution of the disorder problem when the utility function is $U(x) = ln(x)$ and $U(x) = x$, respectively.

### 3.2.3. Optimal Prediction

Take selling stocks as an example, investors aim to sell at the highest point, and the problem is to predict the time of reaching the highest point. Take Process 1 as an example below, the decision maker needs to decide before time 1:

$$V = \inf_{\tau \in [0,1]} \mathbf{E} \left( B_\tau - \max_{0 \leq t \leq 1} B_t \right)^2 \tag{6}$$

The solution to the optimal prediction of a drifted Brownian motion is given by du Toit and Peskir (2007). Investors who prefer technical analysis make decisions based on resistance and support levels. The proposed optimal prediction model is applied not only for predicting the highest point, but also for predicting the resistance and support levels (for details, see Angelis and Peskir (2016)):

$$V = \inf_{\tau \in [0,T]} \mathbf{E}|B_\tau - \mathcal{L}| \tag{7}$$

where $\mathcal{L}$ is a random variable representing a support line or a resistance line with a given distribution.

### 3.2.4. Buy-Low and Sell-High

Assume that an investor is holding an asset, and a simple target is to find the time when the discounted asset payoff reaches the highest point to sell. Following Van Khanh (2012), the utility function is shown as:

$$V(x) = \sup_{0 \leq \tau \leq T} \mathbf{E} \left[ e^{-r\tau} X_\tau \right] \tag{8}$$

Oppositely, if an investor intends to buy an asset, a simple target is to estimate the time when the discounted asset payoff reaches the lowest point to buy. Following Van Khanh (2014), the utility function is shown as:

$$V(x) = \inf_{0 \leq \tau \leq T} \mathbf{E} \left[ e^{-r\tau} X_\tau \right] \tag{9}$$

In a market with trading costs, the investors may need to pay the cost K to sell the stock, which is the problem solved in Guo and Zhang (2005):

$$V(x) = \sup_{0 \leq \tau \leq T} \mathbf{E} \left[ e^{-r\tau} (X_\tau - K) \right] \tag{10}$$

Shiryaev et al. (2008) considered the problem when investors aim to sell at the relative highest price, and the comparison benchmark is the maximum value of price process before selling:

$$V(x) = \inf_{\tau \in [0,T]} \mathbf{E}[1 - \frac{X_\tau}{S_T}] \tag{11}$$

The objective function of du Toit and Peskir (2009) is similar:

$$V(x) = \inf_{0 \leq \tau \leq T} \mathbf{E}[\frac{S_T}{X_\tau}] \tag{12}$$

Dai and Zhong (2012) replaced the above benchmark $S_T$ with the average of price $A_T$:

$$A_T = \begin{cases} \exp\left(\frac{1}{T} \int_0^T \log X_v dv\right), \text{ Geometric average} \\ \frac{1}{T} \int_0^T X_v dv, \text{Arithmetic average} \end{cases} \tag{13}$$

### 3.2.5. Option Pricing

Table 2 shows different types of options. The American option is the most common; the holder of the option can sell/buy the asset at the agreed price at any time before the expiration of the option. The Russian option buyer receives the maximum price of the stock before the strike, reducing the regret of missing the sale peak. The Asian option takes the average of the pre-strike price as the strike price. The British option is similar to the American option of the European option, and the holder can re-price the European option according to the new drift term during the holding period. A bottleneck option can be considered a broader form of Russian option, which is named because its unexercised area resembles a bottleneck. Option 6 is a lookback option, which is designed for hedging the investor's drawdown risk. Option 7 is created for fund managers whose performance fees are linked with the value of the fund exceeding a "high watermark" (historical maximum price). Option 9 is designed to reduce the loss of investors who pursue rises and kill falls. The last option is that the return of the option is related to the state, which is closely related to the regime-switching model.

**Table 2.** Different types of options.

| Label | Objective Function | Note | References (Part) |
|---|---|---|---|
| 1 | $V(x) = \sup_{0 \le \tau \le T} \mathbf{E}\left[e^{-r\tau}(X_\tau - K)^+\right]$ | American Option | Peskir (2005a) |
| 2 | $V(x,s) = \sup_{0 \le \tau \le T} \mathbf{E}\left(e^{-r\tau} S_\tau\right)$ | Russian Option | Shepp and Shiryaev (1993), Peskir (2005b) |
| 3 | $V(x,t) =$ <br> $\sup_{\tau \in [t,T]} \mathbf{E}_t^Q\left\{e^{-r(\tau-t)}\left[\pm(X_\tau - A_\tau)\right]^+\right\}$ | Asian Option | Hansen and Jørgensen (2000) |
| 4 | $V(x,t) =$ <br> $\sup_{0 \le \tau \le T} \mathbf{E}\left[e^{-r\tau}\mathbf{E}^{\mu_c}\left((K - X_T)^+ \mid \mathcal{F}_\tau\right)\right]$ | British Option | Peskir and Samee (2011) |
| 5 | $V(x,s) = \sup_{\tau \ge 0} \mathbf{E}[e^{-r\tau}(S_\tau \wedge C - KX_\tau)^+]$ | Bottleneck option | Ott (2014) |
| 6 | $V(x,s,y) =$ <br> $\sup_{\tau \ge 0} \mathbf{E}_{x,s,y}\left[e^{-r\tau}(K - S_\tau + Y_\tau)^+\right]$ <br> $V(x,s,y) =$ <br> $\sup_{\tau \ge 0} \mathbf{E}_{x,s,y}\left[e^{-r\tau}(KX_\tau - S_\tau + Y_\tau)^+\right]$ | Lookback option | Gapeev and Rodosthenous (2016) |
| 7 | $V(x,s) =$ <br> $\sup_{\tau \ge 0} \mathbf{E}\left[e^{-r\tau}\left(\frac{S_\tau^b}{X_\tau^a} - K\right)^+ \mathbf{1}_{\{\tau < \infty\}}\right]$ <br> $V(x,s) =$ <br> $\sup_{\tau \ge 0} \mathbf{E}\left[e^{-r\tau}\left(S_\tau^b - KX_\tau^a\right)^+ \mathbf{1}_{\{\tau < \infty\}}\right]$ | Watermark option | Rodosthenous and Zervos (2017) |
| 8 | $V(x,m;K) = \sup_{\tau \ge 0} \mathbf{E}\left[e^{-r\tau}(K - m_\tau)^+\right]$ <br> $(m_t = (\inf_{0 \le u \le t} X_u) \wedge m)$ | Lookback option | Woo and Choe (2020) |
| 9 | $V(x,m,s) = \sup_{\tau \ge 0} \mathbf{E}\left[e^{-r\tau}(S_\tau - m_\tau)^+\right]$ | Lookback option | Woo and Choe (2020) |
| 10 | $V_i(x,s) =$ <br> $\sup_{\tau \ge 0} \mathbf{E}[e^{-r\tau}((1 - \Theta_\tau)(L_0 S_\tau - K_0) +$ <br> $\Theta_\tau(L_1 S_\tau - K_1))]$ | $\Theta_\tau$ stands for 0-1 two states | Gapeev et al. (2021) |

### 3.2.6. Other Problems

Battauz et al. (2012) discussed the problem of real options, assuming $(X_t, I_t)$ is a two-dimensional geometric Brownian motion, representing the costs and benefits of the project:

$$V = \sup_{\tau \ge 0} \mathbf{E}\left[e^{-\rho\tau}(X_\tau - I_\tau)^+ \mid \mathcal{F}_0\right] \tag{14}$$

De Angelis and Stabile (2019) studied when investors convert financial assets into annuity insurance. Before the conversion, the income comes from the financial market, and after the conversion, it comes from a fixed amount of annuity every year until death:

$$V_t = \sup_{\tau \in [t,T]} \mathbf{E}[\int_t^{\Gamma_D \wedge \tau} e^{-\rho s} \alpha X_s ds + \mathbf{1}_{\{\Gamma_D \leq \tau\}} e^{-\rho \Gamma_D} X_{\Gamma_D} + P_{\eta+\tau} \int_{\Gamma_D \wedge \tau}^{\Gamma_D} e^{-\rho s} ds \mid \mathcal{F}_t \cap \{\Gamma_D > t\}] \tag{15}$$

Three classical objective functions are summarized in Guerra et al. (2021):

$$V_1(x) = \sup_{\tau \geq 0} \mathbf{E}_x \left[ \int_0^\tau e^{-rs} R(X(s)) ds - e^{-r\tau} K \right] \tag{16}$$

$$V_2(x) = \sup_{\tau \geq 0} \mathbf{E}_x \left[ \int_\tau^{+\infty} e^{-rs} R(X(s)) ds - e^{-r\tau} K \right] \tag{17}$$

$$V_3(x) = \sup_{\tau \geq 0} \mathbf{E}_x \left[ \int_0^\tau e^{-rs} R_1(X_s) ds - e^{-r\tau} K + \int_\tau^\infty e^{-rs} R_2(X_s) ds \right] \tag{18}$$

The three types of functions all consider the case of continuous income. The first function is to solve the problem of when to close a company whose debt is K and cash flow is R, the second one is to solve the problem of when to invest in a new company or project, and the third one is the combination of the two types, which is to solve the problem of when to replace the old one with a new one. Xu (2021) applied the third objective function to investigate the optimal time to intervene to prevent the extinction of endangered species.

## 4. Strategies Based on the Optimal Stopping Method

### 4.1. Sequential Analysis

The drift term of the price process has two alternative values. Investors need to identify which drift term the price process conforms to according to the historical price observed. The solution is given by Van Khanh (2012). The price process $X_t$ obeys the geometric Brownian motion, where $B_t$ is a standard Brownian motion:

$$dX_t = a_t X_t \, dt + \sigma X_t \, dB_t, \quad a_t \in \{a_h, a_l\}, a_h > r > a_l \tag{19}$$

In this situation, the investor's goal is to sell the asset as high as possible, and the investor has a time limit, that is, to sell the asset before T, and the investor's problem is:

$$V_1 = \sup_{0 \leq \tau \leq T} \mathbf{E}\left[e^{-r\tau} X_\tau\right] \tag{20}$$

The initial and posterior distributions of $a_t$ are as follows:

$$\mathbf{P}(a = a_h) = \pi_0; \mathbf{P}(a = a_l) = 1 - \pi_0$$
$$\pi_t = \mathbf{P}\left\{a = a_h \mid \mathcal{F}_t^X\right\} \tag{21}$$

The price process $X_t$ and the posterior probability $\pi_t$ satisfy the following system of stochastic differential equations (SDE):

$$\begin{cases} \frac{dX_t}{X_t} = (a_l + \pi_t(a_h - a_l)) + dt + \sigma d\bar{B}_t \\ d\pi_t = \frac{a_h - a_l}{\sigma} \pi_t(1 - \pi_t) d\bar{B}_t \end{cases} \tag{22}$$

where $\bar{B}$ is a standard Brownian motion under the probability measure $P$, which is defined as:

$$\bar{B}_t = \int_0^t \frac{dX_u - [(1 - \pi_t)a_l - \pi_t a_h] X_u du}{\sigma X_u} \tag{23}$$

Here define a new probability measure $Q$, under which $\{\tilde{B}_t\}$ is a standard Brownian motion:

$$\mathrm{d}\tilde{B}_t = (\omega\pi_t - \sigma)\mathrm{d}t + \mathrm{d}\bar{B}_t \tag{24}$$

Define $\pi_t = \frac{\Phi_t}{\Phi_t+1}$; $\omega = \frac{a_h-a_l}{\sigma}$, then:

$$\eta_t = \exp\left\{-\frac{1}{2}\int_0^t (\sigma - \omega\pi_s)^2 \, \mathrm{d}s + \int_0^t (\omega\pi_s - \sigma)\mathrm{d}\tilde{B}_s\right\} \text{ is the likelihood process}$$

In order to link the objective function with the prior probability, we have the following transformation:

$$\mathbf{E}^P\left[e^{-r\tau}X_\tau\right] = \mathbf{E}^Q\left[e^{-\tau r}\eta_\tau X_\tau\right] = \frac{X_0}{1+\Phi_0}\mathbf{E}^Q\left[e^{(a_l-r)\tau}(1+\Phi_\tau)\right] \tag{25}$$

Recall Equation (20), we have the expectation of the discounted payoff under the risk-neutral measure as:

$$V(t,x) = \sup_{t\leq\tau\leq T}\mathbf{E}\left[e^{-r(\tau-t)}X_\tau|X_t = x\right] = \sup_{t\leq\tau\leq T}\frac{X_t}{1+\Phi_t}\mathbf{E}^Q\left[e^{(a_l-r)(\tau-t)}(1+\Phi_\tau)|\mathcal{F}_t\right] \tag{26}$$

$\Phi_t$ satisfies the following stochastic differential equation:

$$\frac{\mathrm{d}\Phi_t}{\Phi_t} = \omega\sigma\mathrm{d}t + \omega\mathrm{d}\tilde{B}_t \tag{27}$$

The optimal time ($t = 0$) for investors to sell is given by:

$$\tau_* = \inf\{0 \leq u \leq T : \Phi_u \leq b(u)\} \tag{28}$$

where $b(t)$ satisfies:

$$1 + b(t) = e^{(a_l-r)(T-t)} + b(t)\cdot e^{(a_h-r)(T-t)} - \int_0^{T-t}\left\{(a_l-r)e^{(a_l-r)u}F\left(\frac{1}{\omega\sqrt{u}}\left[\ln\frac{(t+u)}{b(t)} - \omega\sigma u + \frac{\omega^2 u}{2}\right]\right)\right.$$
$$\left. + b(t)(a_h-r)e^{(a_h-r)u}F\left(\frac{1}{\omega\sqrt{u}}\left[\ln\frac{b(t+u)}{b(t)} - \omega\sigma u - \frac{\omega^2 u}{2}\right]\right)\right\}\mathrm{d}u \tag{29}$$

where $F(x) = \frac{1}{\sqrt{2\pi}}\int_{-\infty}^x e^{-y^2/2}\,\mathrm{d}y$

The boundary can be converted to the function of $X_t$ using the following formula:

$$X_t = X_0 e^{\varepsilon t}\left(\frac{\Phi_t}{\Phi_0}\right)^\alpha \tag{30}$$

$$\text{where } \alpha = \frac{\sigma}{\omega} = \frac{\sigma^2}{a_h - a_l} \text{ and } \varepsilon = \frac{a_h + a_l - \sigma^2}{2}$$

The ultimate investment strategy is to sell the stock when the price drops to the boundary:

$$\tau^* = \inf\left\{t : X_t \leq \frac{X_0}{\Phi_0^\alpha}e^{\varepsilon t}\cdot b^\alpha(t)\right\} \wedge T \tag{31}$$

### 4.2. Disorder Problem

#### 4.2.1. Case 1: The 'Alarm' Time

The price of an asset is characterized by a geometric Brownian motion, and the drift term of the price process at the beginning is $\mu_1$, at an unobservable random time $\theta$, the drift goes from $\mu_1$ to $\mu_2$,

$$\mathrm{d}X_t = X_t\left(\mu_2 + (\mu_1 - \mu_2)\mathbf{1}_{\{t\leq\theta\}}\right)\mathrm{d}t + \sigma X_t\,\mathrm{d}B_t \tag{32}$$

The prior distribution of $\tau$ is:

$$\mathbf{P}[\theta > t] = (1-p)e^{-\lambda t}, \quad \forall 0 \leq t < \infty \tag{33}$$

where $\mathbf{P}[\theta = 0] = p$. When the drift term of the price process changes, the market moves from one state to another, such as from bull to bear ($\mu_1 > \mu_2$), investors need to recognize market changes as quickly as possible and sell assets in time.

Define the posterior distribution of $\tau$:

$$\pi_t = P\left\{\theta \leq t \mid \mathcal{F}_t^X\right\} \tag{34}$$

Define exponential likelihood-ratio process $L_t$ as:

$$L_t = \frac{\pi_t}{1-\pi_t} = \left(\frac{X_t}{X_0}\right)^{\frac{\mu_2-\mu_1}{\sigma^2}} \exp\left\{-\frac{1}{2\sigma^2}\left((\mu_2-\mu_1)^2 + 2(\mu_2-\mu_1)\left(\mu_1 - \frac{\sigma^2}{2}\right)\right)t\right\} \tag{35}$$

According to the maximum likelihood estimation: $\pi_t = \dfrac{pL_t + (1-p)\int_0^t \lambda e^{-\lambda s}(L_t/L_s)ds}{pL_t + (1-p)\int_0^t \lambda e^{-\lambda s}(L_t/L_s)ds + (1-p)e^{-\lambda t}}$

Equation (32) is transformed as follows:

$$\begin{cases} \frac{\mathrm{d}X_t}{X_t} = (\mu_1 + \pi_t(\mu_2 - \mu_1))\mathrm{d}t + \sigma\mathrm{d}\bar{B}_t \\ \mathrm{d}\pi_t = \lambda(1-\pi_t)\mathrm{d}t + \pi_t(1-\pi_t)\left(\frac{\mu_2-\mu_1}{\sigma}\right)\mathrm{d}\bar{B}_t \end{cases} \tag{36}$$

$$\mathrm{d}\bar{B}_t = \frac{\mathrm{d}X_t - [(1-\pi_t)\mu_1 + \pi_t\mu_2]X_t \, \mathrm{d}t}{\sigma X_t} \tag{37}$$

[Blanchet-Scalliet et al. (2007)](#) referred to two kinds of objective functions used for testing: one is given by Karatzas, another is given by Shiryaev.

For the objective function given by Karatzas:

$$\mathcal{R}(p) = \inf_{\tau \geq 0} \mathbf{E}|\theta - \tau| \tag{38}$$

$$\mathcal{R}(\tau) = \mathbf{E}\left[(\tau-\theta)^+ + (\theta-\tau)^+\right] = \mathbf{E}[(\tau-\theta)^+] + \mathbf{E}[\theta] - \mathbf{E}[\tau \wedge \theta] \tag{39}$$

which is equivalent to:

$$\begin{aligned} \mathcal{R}(\tau) - \mathbf{E}(\theta) &= \mathbf{E}\left[\int_0^\tau \mathbf{1}_{\{\theta \leq t\}}dt - \int_0^\tau \mathbf{1}_{\{\theta > t\}}dt\right] \\ &= \mathbf{E}\int_0^\infty \left(2 \cdot \mathbf{1}_{\{\theta \leq t\}} - 1\right)\mathbf{1}_{\{\theta > t\}}dt = 2 \cdot \mathbf{E}\int_0^\tau \left(\pi_t - \frac{1}{2}\right)dt, \end{aligned} \tag{40}$$

The optimal stopping time of Equation (38) is solved as:

$$\tau^* = \inf\{t \geq 0 \mid \pi_t \geq p^*\} \tag{41}$$

$p^*$ is the unique solution of the following equation in $\left(\frac{1}{2}, 1\right)$.

$$\int_0^{1/2} \frac{(1-2s)e^{-\beta/s}}{(1-s)^{2+\beta}}s^{-2+\beta}ds = \int_{1/2}^{p^*} \frac{(2s-1)e^{-\beta/s}}{(1-s)^{2+\beta}}s^{-2+\beta}ds \tag{42}$$

where $\beta = 2\lambda\sigma^2/(\mu_2-\mu_1)^2$

As for the objective function given by Shiryaev:

$$V(p) = \inf_{\tau \geq 0}\left\{\mathbf{P}(\tau < \theta) + c\mathbf{E}[(\tau-\theta)^+]\right\} = \inf_{\tau \geq 0} \mathbf{E}[1 - \pi_\tau + c\int_0^\tau \pi_t dt] \tag{43}$$

The optimal stopping time is given by:

$$\tau^* = \inf\{t \geqslant 0 \mid \pi_t \geqslant A\} \tag{44}$$

$A$ is the solution of the following equation:

$$
\begin{aligned}
&\int_0^A \exp\left(-\frac{2\lambda\sigma^2}{(\mu_2-\mu_1)^2}\frac{1}{y}\right)\frac{1}{y(1-y)^2}\left(\frac{y}{1-y}\right)^{\frac{2\lambda\sigma^2}{(\mu_2-\mu_1)^2}}\mathrm{d}y \\
&= \frac{(\mu_2-\mu_1)^2}{2\sigma^2 c}\exp\left(-\frac{2\lambda\sigma^2}{(\mu_2-\mu_1)^2}\frac{1}{A}\right)\left(\frac{A}{1-A}\right)^{\frac{2\lambda\sigma^2}{(\mu_2-\mu_1)^2}}
\end{aligned}
\tag{45}
$$

Although the forms of these two objective functions are different, the optimal strategies obtained are similar. Both of them make decisions when the posterior probability that the change takes palace at the current time is greater than the critical value. The posterior probability is calculated according to the likelihood ratio process $L_t$, which is obtained from the observed stock price and other parameters. This strategy does not involve the investor's utility function, but only tells the investor when the market changes.

### 4.2.2. Case 2: Utility Maximization

In Van Khanh (2014), Khanh analyzed the optimal time to sell assets under the framework of the disorder problem according to the maximization of investors' utility.

Consider that $a_t$, $t \geq 0$ is a Markov chain with two states: 0 and 1. $\mathbf{P}(a = a_h) = \pi_0$; $\mathbf{P}(a = a_l) = 1 - \pi_0$, assume that $a_t$ can only go from state 0 to state 1. The transition density function of $a_t$ is: $Q = \begin{bmatrix} -\lambda & \lambda \\ 0 & 0 \end{bmatrix} (\lambda > 0)$, the price satisfies the following SDE:

$$\mathrm{d}X_t = a_t X_t\,\mathrm{d}t + \sigma X_t\,\mathrm{d}B_t, \tag{46}$$

In this case, the investor's goal is to sell the asset as high as possible without a time limit and the investor's problem is

$$V = \sup_{\tau \geq 0} \mathbf{E}\left[e^{-r\tau}X_\tau\right] \tag{47}$$

For $t > 0$, define the posterior distribution $\pi_t = \mathbf{P}\{a = a_h \mid \mathcal{F}_t^X\}$. Recall Equation (36), Equation (46) is similarly transformed:

$$
\begin{cases}
\frac{\mathrm{d}X_t}{X_t} = (a_l + \pi_t(a_h - a_l))\mathrm{d}t + \sigma\mathrm{d}\bar{B}_t \\
\mathrm{d}\pi_t = \lambda_1(1 - \pi_t)\mathrm{d}t + \pi_t(1 - \pi_t)\left(\frac{a_h - a_l}{\sigma}\right)\mathrm{d}\bar{B}_t
\end{cases}
\tag{48}
$$

The definitions of $\Phi_t$, $\omega$, $(\bar{B}_t, P)$, $(\tilde{B}_t, Q)$ and $\eta_t$ are the same as in the case of sequential analysis (see Equation (23) and the subsequent notations), then the following equations hold:

$$\mathbf{E}^P[e^{-r\tau}X_\tau] = \mathbf{E}^Q[\eta_\tau e^{-r\tau}X_\tau] = \frac{X_0}{1+\Phi_0}\mathbf{E}^Q[e^{(a_l-\lambda-r)\tau}(1+\Phi_\tau)] \tag{49}$$

Recall Equation (47), now we have:

$$V = \sup_{\tau \geq 0}\frac{X_0}{1+\Phi_0}\mathbf{E}^Q[e^{(a_l-\lambda-r)\tau}(1+\Phi_\tau)] \tag{50}$$

where $\Phi_t$ satisfies:

$$\mathrm{d}\Phi_t = (\lambda + (\lambda+\sigma\omega)\Phi_t)\mathrm{d}t + \omega\Phi_t\,\mathrm{d}\tilde{B}_t \tag{51}$$

The optimal stopping time of investors is given by:

$$\tau^* = \inf\{t \geq 0 : \Phi_t \geq A\} = \inf\left\{t \geq 0 : \pi_t \geq \frac{A}{A+1}\right\} \tag{52}$$

where $A$ is the solution of the following equation:

$$2\int_0^\infty e^{-\alpha u} u^{(\beta+\gamma-3)/2}(1+Au)^{(\gamma-\beta+1)/2}\,du = (1+A)(\gamma-\beta+1)\int_0^\infty e^{-\alpha u}u^{(\beta+\gamma-1)/2}(1+Au)^{(\gamma-\beta-1)/2}\,dt \tag{53}$$

$$\alpha = \frac{2\lambda}{\omega^2} > 0, \beta = \alpha + \frac{2\sigma}{\omega} > 0, \gamma = \sqrt{(\beta-1)^2 + \frac{8(\lambda+r-a_l)}{\omega^2}} > |\beta-1| \tag{54}$$

As before, the best time for investors to sell is when they have enough confidence that the market has turned from bull to bear, i.e., the posteriori probability is greater than the critical value.

### 4.3. Optimal Prediction

An important question in the asset pricing area is to find their resistance and support levels when investors conduct the technical analysis. The game between buyers (demand) and sellers (supply) often leads to an upward or downward trend in asset prices. The resistance (support) line is the level at which most traders are willing to sell (or buy) an asset. When the levels are reached, prices either fall for a while (reaching the resistance line) or rise (reaching the support line). Therefore, the resistance and support levels can be used to construct effective trading strategies and gain extra earnings. Angelis and Peskir (2016) presents a prediction method of the support level and resistance level based on the optimal stopping model, which is shown as follows:

Suppose that the price of assets follows geometric Brownian motion: $dX_t = \mu X_t dt + \sigma X_t dB_t, X_0 = x$, the support line/resistance line $\ell$ is a random variable with a distribution function of $F$. Assume that $F$ and $X_t$ are independent, this assumption is very reasonable because both the support line and the resistance line depend on the information of the past (before time 0) and have nothing to do with the future. When $\mu > 0$, investors look for the resistance line to determine the best time to sell assets. When $\mu < 0$, investors look for the support line to determine the best time to buy assets. The investment decision is completed before t. The objective function of the investor is:

$$V(x) = \inf_{0 \leq \tau \leq T} \mathbf{E}[|X_\tau^x - \ell|] \tag{55}$$

$$\mathbf{E}[|x - \ell|] = 2\int_0^x \left(F(y) - \frac{1}{2}\right)dy + \mathbf{E}\ell \tag{56}$$

Set $G(x) = \int_0^x \left(F(y) - \frac{1}{2}\right)dy$, the problem can be converted to finding $\tau$ so that:

$$V(x) = \inf_{0 \leq \tau \leq T} \mathbf{E}[G(X_\tau^x)] \tag{57}$$

To describe the solution of the problem, first define the following equation:

$$J(t,x) = \mathbf{E}_x[G(X_{T-t})] = \int_0^\infty G(z)f(T-t,x,z)dz$$

$$H(x) = \mu x\left(F(x) - \frac{1}{2}\right) + \frac{\sigma^2}{2}x^2 F'(x)$$

$$K(s,x,y) = \mathbf{E}_x\left[H(X_s)\mathbf{1}_{\{X_s>y\}}\right] = \int_y^\infty H(z)f(s,x,z)dz \tag{58}$$

$$L(s,x,y) = \mathbf{E}_x\left[H(Y_s)\mathbf{1}_{\{X_s<y\}}\right] = \int_0^y H(z)f(s,x,z)dz$$

The optimal stopping time is:

$$
\tau^* = \begin{cases} \inf\{t \in [0,T] \mid X_t \geq b_1(t)\}, & \text{when } \mu > 0 \\ 0, & \text{when } \mu = 0, \\ \inf\{t \in [0,T] \mid X_t \leq b_2(t)\}, & \text{when } \mu < 0, \end{cases} \tag{59}
$$

$b_1(t)$ and $b_2(t)$ are solutions to the following equations, respectively:

$$
J(t, b_1(t)) = G(b_1(t)) + \int_t^T K(s - t, b_1(t), b_1(s)) ds \tag{60}
$$

$$
J(t, b_2(t)) = G(b_2(t)) + \int_t^T L(s - t, b_2(t), b_2(s)) ds \tag{61}
$$

As for the distribution F of the support line and resistance line, two examples are given:

① Both the support line and the resistance line follow an exponential distribution, that is for $x > 0$, then $F'(x) = \lambda e^{-\lambda x}$; for $x \leq 0$ then $F'(x) = 0$.

② The distribution F of the support line is the same as the maximum process of $X_t$, that is

$$
S = \sup_{t \geq 0} \left[ y \exp\left( \sigma B_t + \left( \mu - \left( \sigma^2/2 \right) \right) t \right) \right] \tag{62}
$$

The analytic form of the distribution function of S can be solved:

$$
F(x) = 1 - \left( \frac{y}{x} \right)^{1 - \mu/\left( \sigma^2/2 \right)} \tag{63}
$$

du Toit and Peskir (2007) is concerned with the prediction of the drifted Brownian motion in the finite time: $B^\mu = \left( B_t^\mu \right)_{0 \leq t \leq T}$ is a drifted Brownian motion with a drift $\mu \in \mathbb{R}$, $S_t^\mu = \max_{0 \leq s \leq t} B_s^\mu$. Consider the optimal prediction problem in the finite period:

$$
V = \inf_{0 \leq \tau \leq T} \mathbf{E}\left[ \left( B_\tau^\mu - S_T^\mu \right)^2 \right] \tag{64}
$$

By transforming the above problem into a free boundary problem, we can obtain:

$$
\tau_* = \inf\left\{ t_* \leq t \leq T \mid b_1(t) \leq S_t^\mu - B_t^\mu \leq b_2(t) \right\} \tag{65}
$$

In the case of $\mu > 0$, when t is in $[t_*, T](t_* \in [0, T])$, $b_1(t)$ is a decreasing function, $b_2(t)$ an increasing function. In the case of $\mu \leq 0$, $b_2(t)$ is infinite, $b_1(t)$ first increases and then decreases at [0,T]. For more details of $b_1(t)$ and $b_2(t)$, see du Toit and Peskir (2007).

*4.4. Buy-Low and Sell-High*

4.4.1. Case 1: Sell-High

The asset price follows the geometric Brownian motion: $dX_t = \mu X_t dt + \sigma X_t dB_t$, according to Shiryaev et al. (2008), consider the discounted asset price $dP_t = (\mu - r)P_t dt + \sigma P_t dB_t$. The investor buys the stock at time 0, and for some reason, they have to sell the stock by a pre-specified date T. The problem for the investor is to sell the stock as high as possible.

$$
V_1 = \max_{\tau \in [0,T]} \mathbf{E}[U(\frac{P_\tau}{M_T})] \qquad M_t = \max_{s \in [0,t]} P_s \tag{66}
$$

When $U(x) = x^\gamma$, the problem is equivalent to $\inf_{\tau \in [0,T]} \mathbf{E}[1 - \frac{P_\tau}{M_T}]$, which means that the investor wishes to minimize the expected relative error between the (discounted) selling price and the (discounted) highest price on [0, T]. The above problem can be further

simplified as $\sup_{\tau \in [0,T]} \mathbf{E}[\frac{P_\tau}{M_\tau}]$. The solution to the problem is a sample bang–bang solution, which is:

$$
\tau^* = \begin{cases} T, & \text{if } \mu - r - \frac{\sigma^2}{2} > 0 \\ \text{any time between } [0,T], & \text{if } \mu - r - \frac{\sigma^2}{2} = 0 \\ 0, & \text{if } \mu - r - \frac{\sigma^2}{2} < 0 \end{cases} \tag{67}
$$

du Toit and Peskir (2009) presented another description of the problem; recall that $S_t$ is the maximum process of $X_t$:

$$
V_2 = \inf_{0 \le \tau \le T} \mathbf{E}\left(\frac{S_T}{X_\tau}\right) = \inf_{0 \le \tau \le T} \mathbf{E}\left(e^{\sigma\left(S_T^\lambda - B_\tau^\lambda\right)}\right) \tag{68}
$$

$$
B_t^\lambda = B_t + \lambda t, \quad \lambda = (\mu - r - \frac{\sigma^2}{2})/\sigma^2, \quad S_t^\lambda = \max_{s \in [0,t]} B_s^\lambda \tag{69}
$$

Defining $Z_t = S_t^\lambda - B_t^\lambda$, the two-dimensional problem of $(S_t^\lambda, B_t^\lambda)$ is transformed into a one-dimensional problem of $Z_t$.

Define $G(t,z) = \mathbf{E}[e^{\sigma\left(z \vee S_{T-t}^\lambda\right)}]; \quad H(t,z) = G_t - \lambda G_z + 1/2 G_{zz}$
$J(t,z) = \mathbf{E}_{t,z}[G(T, Z_T)]; \quad K(t,z,r,y) = \mathbf{E}_{t,z}[H(t+r, Z_{t+r})\mathbf{1}_{\{Z_{t+r} > y\}}]$
Recalling Equation (68), the optimal stopping model becomes:

$$
V(t,z) = \inf_{0 \le \tau \le T-t} \mathbf{E}[(G(t+\tau, Z_{t+\tau})|Z_t = z)] \tag{70}
$$

The optimal stopping time is given by:

$$
\tau^* = \begin{cases} 0, & \text{when } \mu - r \le 0 \\ \inf_{\tau \in [0,T]}(S_\tau^\lambda - B_\tau^\lambda \ge b(\tau)), & \text{when } \mu - r \in (0, \sigma^2) \\ T, & \text{when } \mu - r \ge \sigma^2 \end{cases} \tag{71}
$$

where $b(t)$ is the unique solution of the following nonlinear Volterra integral equation.

$$
J(t, b(t)) = G(t, b(t)) + \int_0^{T-t} K(t, b(t), s, b(t+s))ds \tag{72}
$$

$$
V_2 = V(0,0) \tag{73}
$$

### 4.4.2. Case 2: Buy-Low

Liu et al. (2020) considered a more complex utility function to describe the asset purchase by investors where the optimal stopping model is firstly simplified and then solved by numerical method. The asset price follows the geometric Brownian motion: $dX_t = \mu X_t dt + \sigma X_t dB_t$; consider such a optimal stopping model:

$$
V(t,x) := \sup_{t \le \tau \le T} \mathbf{E}\left[(K - X_\tau)\mathbf{1}_{\{S_{\tau,T} > K\}} \mid X_t = x\right] \tag{74}
$$

$$
S_{r,s} := \sup_{r \le v \le s} X_v \tag{75}
$$

The objective function describes a scenario in which the investor aims to determine an optimal time to buy in to enlarge the profits and to ensure the deal closed by hitting the take-profit level before time $T$. Because the investor thinks that the profitable level $K$ is

only possible before the time $T$, and after time $T$ the price is unpredictable and riskier, so the deal is closed before $T$. Equation (74) is simplified as follows:

$$
\begin{aligned}
V(t, x) &= \sup_{t \leq \tau \leq T} \mathbf{E}\Big[\mathbf{E}\big[(K - X_\tau)\mathbf{1}_{\{S_{\tau,T} > K\}} \mid X_\tau, X_t = x\big] \mid X_t = x\Big] \\
&= \sup_{t \leq \tau \leq T} \mathbf{E}\Big[(K - X_\tau)\mathbf{E}\big[\mathbf{1}_{\{S_{\tau,T} > K\}} \mid X_\tau\big] \mid X_t = x\Big] \\
&= \sup_{t \leq \tau \leq T} \mathbf{E}[G(\tau, X_\tau) \mid X_t = x]
\end{aligned}
\tag{76}
$$

$$
G(t, x) = (K - x)P\left(S^\lambda_{T-t} > \frac{1}{\sigma}\log\left(\frac{K}{x}\right)\right)
\tag{77}
$$

where

$$
\lambda := \frac{\mu}{\sigma} - \frac{\sigma}{2}, \quad B^\lambda_s := B_s + \lambda s, \quad S^\lambda_t := \max_{0 \leq s \leq t} B^\lambda_s
\tag{78}
$$

In the case of $\mu > 0$, it is proved that the best strategy is to buy immediately, that is $\tau^* = t$; in the case of $\mu < 0$, due to the complexity of this case, an iterative numerical algorithm is presented without an analytical solution, and examples are given.

### 4.5. Regime-Switching

In the real financial market, the asset price would not always stay in a fixed upward or downward trend, but would appear alternately in the combined form of the bull market, bear market, and consolidation period. therefore, the asset process cannot be well described by a simple geometric Brownian motion. The regime-switching model introduces a Markov chain that takes values in several states. The drift parameter and diffusion parameter of the price process take different values in different states, which can better describe the real financial market.

Assume that the price of the asset is $X_t$, a regime-switching model is described as follows:

$$
dX_t = X_t \mu_{\epsilon(t)}dt + X_t \sigma_{\epsilon(t)}dB_t, \quad X_0 = x
\tag{79}
$$

where $\epsilon(t) \in \{1, 2, \ldots, S\}$ is a finite-state continuous-time Markov chain, $\epsilon(t)$ is generated by $A$.

$$
A = \begin{pmatrix}
a_{1,1} & a_{1,2} & \cdots & a_{1,S} \\
a_{2,1} & a_{2,2} & \cdots & a_{2,S} \\
\vdots & \vdots & \ddots & \vdots \\
a_{S,1} & a_{S,2} & \cdots & a_{S,S}
\end{pmatrix}
\tag{80}
$$

where $a_{i,i} = -\sum_{k=0, k \neq i}^{S} a_{i,k}, i = 1, 2, \ldots, S$. The parameters $a_{i,k}, i \neq k$ which are positive, define the probability of the jump process of $\epsilon(t)$ in a small interval of time. If at time t, $\epsilon(t) = i$, then with probability $a_{i,j}\Delta t$, $\epsilon(t)$ will transit from state $i$ to state $j$ at $t + \Delta t$, and the probability of remaining in the state $i$ is $1 - \sum_{k=1, k \neq i}^{S} a_{i,k}\Delta t = 1 + a_{i,i}\Delta t$.

Consider a two-dimensional case in which the market has two states, corresponding to a bear market and a bull market.

$$
\mu_{\epsilon(t)} = \begin{cases} \mu_1, & \epsilon(t) = 1, \\ \mu_2, & \epsilon(t) = 2 \end{cases} \quad
\sigma_{\epsilon(t)} = \begin{cases} \sigma_1, & \epsilon(t) = 1 \\ \sigma_2, & \epsilon(t) = 2 \end{cases} \quad
A = \begin{pmatrix} -\lambda_1 & \lambda_1 \\ \lambda_2 & -\lambda_2 \end{pmatrix}
\tag{81}
$$

Let $\tau_i$ denote the time to leave the state i, then:

$$
\mathbf{P}(\tau_i > t) = e^{-\lambda_i t}, \quad i = 1, 2
\tag{82}
$$

### 4.5.1. An Optimal Solution

Consider an investor who holds a share of stock with the price $X_t$ at time $t$. The investor's goal is to maximize the discounted expected payoff with the amount K to be paid back when the investor sells the stock

$$V_i^*(x) = \sup_{\tau \geq 0} \mathbf{E}\left[e^{-r\tau}(X_\tau - K) \mid X_0 = x, \epsilon(0) = i\right] \tag{83}$$

The problem is solved by Guo and Zhang (2005). To better describe the solution, here, define $B_1 > B_2$ are two roots of the following equation:

$$x^2 - (\mu_1 - \lambda_1 + \mu_2 - \lambda_2)x + (\mu_1 - \lambda_1)(\mu_2 - \lambda_2) - \lambda_1\lambda_2 = 0 \tag{84}$$

① In the case of $r \leq B_1$, $\tau^* = \infty$.

② In the case of $r > max(\mu_1, \mu_2) \geq B_1$. Suppose that $x_i$ is the critical value at which the price process stops immediately at the state i is the optimal decision, when $x \geq x_i$, it is optimal to sell immediately, $x_i$ satisfies:

$$H^j(x_1, x_2, F_1(x_1, \phi(x_1)), F_2(x_2, \phi(x_2))) = 0 \tag{85}$$

when $x_1 < x_2$, then $j = 1$; when $x_1 > x_2$, then $j = 2$. Some other parameters are needed:

$$g_1(\beta)g_2(\beta) = \lambda_1\lambda_2 \quad \beta_1 > \beta_2 > 0 > \beta_3 > \beta_4 \quad \text{are the roots to the equation} \tag{86}$$

$$\text{where} \quad g_i(\beta) = \lambda_i + r - \left(\mu_i - (1/2)\sigma_i^2\right)\beta - (1/2)\sigma_i^2\beta^2, (i = 1, 2) \tag{87}$$

$$l_i^1 = g_1(\beta_i)/\lambda_1 = \lambda_2/g_2(\beta_i), l_i^2 = 1/l_i^1 \text{ for } i = 1, 2, 3, 4. \tag{88}$$

$\gamma_i^1 (i = 1, 2)$ are the real roots to the equation:

$$\mu_2\gamma + \frac{1}{2}\sigma_2^2\gamma(\gamma - 1) = r + \lambda_2 \tag{89}$$

$\gamma_i^2 (i = 1, 2)$ are the real roots to the equation:

$$\mu_1\gamma + \frac{1}{2}\sigma_1^2\gamma(\gamma - 1) = r + \lambda_1 \tag{90}$$

Based on the above notations, the functions used in Equation (85) are defined as follows:

$$\phi(x) = -\frac{\lambda_2 K}{r + \lambda_2} + \frac{\lambda_2 x}{r + \lambda_2 - \mu_2} \tag{91}$$

$$
\begin{aligned}
F_1^j(x, g(x)) = \begin{pmatrix} 1 & 1 \\ \gamma_1 & \gamma_2 \end{pmatrix}^{-1} & \left[ \begin{pmatrix} l_1^j & l_2^j \\ l_1^j\beta_1 & l_2^j\beta_2 \end{pmatrix} \begin{pmatrix} 1 & 1 \\ \beta_1 & \beta_2 \end{pmatrix}^{-1} \right. \\
& \left. \times \begin{pmatrix} x - K \\ x \end{pmatrix} - \begin{pmatrix} g(x) \\ xg'(x) \end{pmatrix} \right]
\end{aligned}
\tag{92}
$$

$$F_2^j(x, g(x)) = \begin{pmatrix} 1 & 1 \\ \gamma_1^j & \gamma_2^j \end{pmatrix}^{-1} \begin{pmatrix} x - K - g(x) \\ x - xg'(x) \end{pmatrix}. \tag{93}$$

$$H^j(x_1, x_2, Y_1, Y_2) = \begin{pmatrix} x_1^{-\gamma_1^j} & 0 \\ 0 & x_1^{-\gamma_2^j} \end{pmatrix} Y_1 - \begin{pmatrix} x_2^{-\gamma_1^j} & 0 \\ 0 & x_2^{-\gamma_2^j} \end{pmatrix} Y_2 \tag{94}$$

③ As for $max(\mu_1, \mu_2) > r > B_1$, the problem is not solved.

The above analytical solution does not include the case where r is between the drift terms of the two states. The unsolved case is common in the real world, so the solution is not very practical.

### 4.5.2. A Suboptimal Solution

To solve the problem in a real world, we consider a suboptimal solution. Zhang (2001) considered a suboptimal solution of the optimal stopping problem under the regime-switching model. Here is a simple strategy where investors implement a stop loss level and a profit level, such as selling the asset when it falls 10%($z_1$) or rise 20%($z_2$), let $X_t = X_0 e^{Z_t}$, $Z_t \in [-z_1, z_2]$, $B_t$ is a standard Brownian motion:

$$dZ_t = \alpha_{\epsilon(t)} dt + \sigma_{\epsilon(t)} dB_t \tag{95}$$

where $\alpha_i = \mu_i - \sigma_i^2/2, i \in \{1, 2, \ldots, S\}$

Let:

$$\tau = \inf\{t \geq 0 : Z_t \notin (-z_1, z_2)\} \tag{96}$$

Given $\epsilon(t) = i$ and $X_0 = x$, $\rho$ (can be estimated) is the discount factor, let $v(x, i)$ denote the objective function:

$$v(i) = \mathbf{E}\big[\Phi(Z_t)e^{-\rho\tau} \mid \epsilon(0) = i\big] \tag{97}$$

The problem for the investor is to choose the best $z_1$ and $z_2$ to maximize the objective function:

$$V = V(z_1, z_2) = \sum_{i=1}^{m} p_i v(i) \tag{98}$$

where $p_i$ is the probability that $\epsilon(0) = i$. Take $\Phi(z) = e^z - 1$ as an example:

$$v(i) = \mathbf{E}\left[\left(\frac{X(\tau) - X_0}{X_0}\right)e^{-\rho\tau} \mid \epsilon(0) = i\right] = \mathbf{E}\big[\Phi(Z_t)e^{-\rho\tau} \mid \epsilon(0) = i\big] \tag{99}$$

In the case of two states, $S = 2$, let $\eta_i$ for i = 1, 2, 3, 4 be the four real roots to make the following function equal to 0:

$$\psi(\eta) = \frac{\sigma_1^2 \sigma_2^2}{4}\left\{\left(\eta^2 + \frac{2\alpha_1}{\sigma_1^2}\eta - \frac{2(\rho + \lambda_1)}{\sigma_1^2}\right)\left(\eta^2 + \frac{2\alpha_2}{\sigma_2^2}\eta - \frac{2(\rho + \lambda_2)}{\sigma_2^2}\right) - \frac{4\lambda_1\lambda_2}{\sigma_1^2\sigma_2^2}\right\} \tag{100}$$

$$\text{Let} \quad \kappa_i = \frac{1}{\lambda_1}\left(-\frac{\sigma_1^2}{2}\eta_i^2 - \alpha_1\eta_i + \rho + \lambda_1\right) \tag{101}$$

$(c_1, c_2, c_3, c_4)$ can be solved from

$$\begin{pmatrix} 1 & 1 & 1 & 1 \\ \kappa_1 & \kappa_2 & \kappa_3 & \kappa_4 \\ e^{\eta_1(z_1+z_2)} & e^{\eta_2(z_1+z_2)} & e^{\eta_3(z_1+z_2)} & e^{\eta_4(z_1+z_2)} \\ \kappa_1 e^{\eta_1(z_1+z_2)} & \kappa_2 e^{\eta_2(z_1+z_2)} & \kappa_3 e^{\eta_3(z_1+z_2)} & \kappa_4 e^{\eta_4(z_1+z_2)} \end{pmatrix}\begin{pmatrix} c_1 \\ c_2 \\ c_3 \\ c_4 \end{pmatrix} = \begin{pmatrix} \Phi(-z_1) \\ \Phi(-z_1) \\ \Phi(z_2) \\ \Phi(z_2) \end{pmatrix} \tag{102}$$

The investor chooses $(z_1, z_2)$ to maximize (This process is achieved by numerical methods)

$$V = p_1 v(1) + p_2 v(2) = \sum_{i=1}^{4} c_i(p_1 + \kappa_i p_2)e^{\eta_i z_1} \tag{103}$$

### 4.5.3. Parameter Estimation

As we mention in the introduction, we focus on the application in financial market because the financial market data are sufficient to estimate the model parameters. Here, we

give an example, which is given by Zhang (2001). Let $X_i, i = 0, 1, \ldots, n$ be the closing price of the stock.

$$\zeta_i = \log X_i - \log X_{i-1}, i = 1, 2, \ldots, n \tag{104}$$

Let $t_i = i/N_0$, $N_0 = 252$ is the number of trading days in a year.

$$\zeta_i = \alpha(t_i - t_{i-1}) + \sigma\left(B_{t_i} - B_{t_{i-1}}\right) \sim N\left(\frac{\alpha}{N_0}, \frac{\sigma^2}{N_0}\right) \tag{105}$$

$$\sigma = \sqrt{N_0} \cdot \sqrt{\frac{1}{n-1} \sum_{i=1}^{n} \left(\zeta_i - \bar{\zeta}\right)^2} \tag{106}$$

The range of asset price in a year is divided into several parts. Each part of assets has a clear upward (state 1) or downward (state 2) trend. Use $X_i^{up}$ and $X_i^{down}$ to represent the asset price in the upward and downward parts, respectively, $n^{up}$ and $n^{down}$ to represent the days of the total upward and downward parts, respectively, $k^{up}$ and $k^{down}$ represent the number of rising and falling regimes, respectively, $N^{up}$ and $N^{down}$ represent total magnitudes of increase and decrease in the two regimes, respectively. The estimate of $\sigma_1$ is obtained by replacing the $X_i$ in $\zeta_i$ with $X_i^{up}$; $\sigma_2$ is estimated similarly. The estimates of $\alpha_i$ and $\lambda_i$ are as follows:

$$
\begin{aligned}
\alpha_1 &= N_0\left(\frac{\log\left(X_0^{up} + N^{up}\right) - \log X_0^{up}}{n^{up}}\right), \\
\alpha_2 &= N_0\left(\frac{\log\left(X_0^{down} - N^{down}\right) - \log X_0^{down}}{n^{down}}\right).
\end{aligned}
\tag{107}
$$

$$\frac{1}{\lambda_1} = \frac{n^{up}}{N_0}\frac{1}{k^{up}}; \quad \frac{1}{\lambda_2} = \frac{n^{down}}{N_0}\frac{1}{k^{down}} \tag{108}$$

### 4.6. Numerical Solutions

The above focuses on the analytical solutions of optimal stopping, as the numerical solutions are less elegant or general than closed form solutions; as such, economists try to avoid numerical methods. However, due to the complexities in the real-world, analytically insoluble models are common in finance, deriving economic insights from a realistic numerical model of an economic system is preferable to deriving irrelevant answers from an unrealistic but solvable model. Relevant papers on numerical solutions are provided here.

Optimal stopping problems are often transformed into variational inequalities. Consider the continuous $d$-dimensional state process $\mathbf{X}$:

$$d\mathbf{X} = \mu(\mathbf{X})dt + \sigma(\mathbf{X})d\mathbf{B} \tag{109}$$

where $\mu$ is a $d \times 1$ drift vector, $\sigma$ is a $d \times d$ diffusion matrix and $\mathbf{B}$ is the $d$-dimensional vector of independent Wiener processes. Balikcioglu et al. (2011) consider the optimal stopping problem:

$$V(\mathbf{X}) = \max_{\tau \geq 0} \mathbf{E}\left[\int_0^{\tau} e^{-rt} f(\mathbf{X}_t)dt + e^{-r\tau} U(\mathbf{X}_\tau) \mid \mathbf{X}_0 = \mathbf{X}\right] \tag{110}$$

which can be transformed to:

$$0 = \min[rV(\mathbf{X}) - \mathcal{L}V(\mathbf{X}) - f(\mathbf{X}), V(\mathbf{X}) - U(\mathbf{X})] \tag{111}$$

where $\mathcal{L}$ is the infinitesimal generator of **X**:

$$\mathcal{L} = \sum_{i=1}^{d} \mu_i(\mathbf{X}) \frac{\partial}{\partial \mathbf{X}_i} + \frac{1}{2} \sum_{i=1}^{d} \sum_{j=1}^{d} \Sigma_{ij}(\mathbf{X}) \frac{\partial^2}{\partial \mathbf{X}_i \partial \mathbf{X}_j}; \quad \Sigma_{ij}(\mathbf{X}) = \sum_{k} \sigma_{ik}(\mathbf{X}) \sigma_{ik}(\mathbf{X}) \tag{112}$$

Balikcioglu et al. (2011) provided a numerical method for soving the variational inequality. The value function V is approximated by piecewise linear functions $\phi(\mathbf{X})c$, where $\phi$ is a set of n basis functions and c is an n-vector of coefficients.

Sims et al. (2018) considered the resource population with demographic and environmental stochasticities, an analytic solution is not possible for the optimal timber rotation problem as the underlying process is complicated. The numerical method used in Balikcioglu et al. (2011) is applied for direct implications.

## 5. Strategy Performance

The previous part summarizes the construction process of the optimal stopping strategy from the theoretical aspect. Practically, a strand of literature applies the optimal stopping method to the financial markets and evaluate how the optimal stopping strategies perform.

Boubaker et al. (2021) incorporate the maximum drawdown into the objective function, and investors weigh the trade-off between tolerating the current drawdown in the hope of a new high price level and selling immediately:

$$g(X_t, S_t) = \phi_1 \sqrt{S_t} - \phi_2 (S_t - X_t) \tag{113}$$

The optimal strategy is given by a boundary X of the drawdown, it is optimal to sell immediately when the current drawdown is greater than X. The strategy can provide an efficient stop-loss signal in several specific periods, such as the 2008 financial crisis and the European debt crisis. In some emerging markets, for example the Chinese stock market, by following the optimal stopping strategy, investors are able to stop their losses when the stock market crashed in the financial crisis. Compared with the maximum drawdown of 71.8%, investors who employ the optimal stopping models only lost 8.7% at the stop-loss point. In the Indian market, the optimal stopping strategy limited the losses under 9.68% and 4.67% during the financial crisis and European debt crisis, respectively. In the Brazilian stock market, investors can use the optimal stopping strategy to stop their losses before the crash of the financial crisis. The drawdown of the stock at the selling time suggested by the strategy was only 3.7% compared with the 60% maximum drawdown. In the Russian stock market, this strategy limited losses to 11.59% during the financial crisis, with the maximum drawdown during that period being 77.91%.

The optimal prediction method is adopted in Boubaker et al. (2021) to carry out risk management of crude oil futures, and the two periods of sharp rise and fall of crude oil market from January 2007 to December 2008 and from February 2009 to October 2011 are selected. Although the selling time suggested by the optimal prediction method lags behind the peak of the price, it can effectively reduce losses. For example, during the financial crisis, the crude oil futures are sold when the drawdown is 28%, followed by a maximum drawdown of 77%.

In Shiryaev et al. (2014), the optimal stopping method which is similar to Case 2 of the disorder problem in Section 4, is applied in two situations: the Apple price bubble, which began on 6 March 2009 at a local low of $82.33, and the Internet technology bubble, which began around 2000 and was measured by the NASDAQ (NDX100), on which futures contracts were sold. The model attempts to predict an exit point near the price peak (or valley, for a short position) and performs well in both situations. In the first situation, the model works equally well for both early and late entering dates, giving nearly 90% of the maximum price on the exiting date. In the second situation, the exit yielded about 75% of

the maximum price, with investor gains of about 40% to 60% a year (about 25% to 45% a year), depending on the long-position(short-position) entry point suggested by the model.

In Liu and Zhan (2022), a filter rule strategy is considered. They assume that the investor's utility function is influenced by both the positive utility of returns and the aversion to drawdown, myopic loss aversion is persistent and can lead to the most profitable opportunities. With the application of the filter rule, the drawdown of investors in the S&P 500 during the 2008 financial crisis can be reduced to $-21.2\%$ and the drawdown of investors in the Chinese stock market in 2015 can be reduced to $-24.8\%$.

## 6. Conclusions

This paper introduces the analysis framework about the optimal stopping model as well as the investment decision-making strategy based on the optimal stopping model. It can be seen that research on the optimal stopping model mainly focuses on three aspects: the first is the selection of the underlying process, from the simple geometric Brownian motion to the regime-switching model, etc. Second, according to a specific problem, choose the appropriate objective function to optimize, which calls for consideration about all factors affecting the objective function, such as the historical maximum, time limit, and others. The specific form of the function and the dimension of the optimal stopping model needs to be seriously considered. Finally, it is about the study of the solution of the optimal stopping model. It is more challenging to obtain the analytical solution given a relatively complex utility function. Relevant literature studies the existence, uniqueness, and general solution of the optimal stopping models.

It can also be found from the previous summary that under the five stop-investing strategies, the strategy of simply buy-low and sell-high is not very practical, but it can be used to explain the view of value investing, that is, to buy good stocks and hold on to them. It is easy to obtain valuable analytical solutions for sequential analysis and disorder problems where the Bayesian approach is used to update beliefs according to new information. As the regime-switching model is very complex, the existing analytical solution of this type of optimal stopping model can not provide a practical strategy, and a practical suboptimal solution is given in the reference. The optimal prediction problem can obtain valuable analytical solutions, but it is not suitable for long-term investment because the stochastic process that the problem is based on is relatively simple. As for the form of a solution, an optimal stopping strategy provides a boundary beyond which an asset needs to be bought/sold immediately.

However, several relevant research problems remain unexplored by the current literature. One potential problem with the existing optimal stopping strategies is that they cannot be used for multi-asset decision-making. The high-dimensional optimal stopping models are too complex to be well solved. Generally speaking, the most commonly used method for high-dimensional problems is to reduce dimension through the change of measure, space, and time. In the case of multi-asset, the dimension can not be reduced. There are few studies on multi-asset decision-making in the existing literature, so this problem needs further consideration.

Another natural extension could be the choice of the objective function in investment decision-making. Objective function needs to be combined with practical problems. Various new options using the optimal stopping method to price will provide ideas for selecting the appropriate objective function of the optimal stopping model. New options include risks such as maximum retracement into the hedging scope, while investors should not only maximize returns but also reduce drawdown risk when making decisions. It is possible to find an appropriate objective function by transforming the pricing of options into investment problems.

Research in behavioral finance tries to improve the psychological realism of the traditional model through more realistic assumptions about individual beliefs, such as the extrapolation framework and the overconfidence framework, and individual preferences, such as the gain–loss utility framework inspired by prospect theory. For optimal stopping

models, individual beliefs are reflected by the underlying processes and individual preferences are reflected by the utility functions. Irrational considerations may improve the performance of the optimal stopping strategy and produce some new implications.

Finally, as a kind of less concerned investment strategy, the investment strategy based on the optimal stopping method provides a new idea for investors to decide their optimal investment time. It has a solid theoretical foundation and provides investors with a reference standard. With the substantial development in the optimization of numerical algorithms and the optimal stopping theory, researchers and practitioners can combine different objective functions with different stochastic processes. Thus, they can put forward new investment strategies based on the proposed optimal stopping methods.

**Author Contributions:** Conceptualization, Z.L.; methodology, Y.M.; formal analysis, Z.L. and Y.M.; investigation, Z.L. and Y.M.; writing—original draft preparation, Y.M.; writing—review and editing, Z.L. and Y.M. All authors have read and agreed to the published version of the manuscript.

**Funding:** This research received no external funding.

**Informed Consent Statement:** Not applicable.

**Data Availability Statement:** Not applicable.

**Conflicts of Interest:** The authors declare no conflict of interest.

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
