# Peer review of "Optimal Stopping Methods for Investment Decisions: A Literature Review"

_ijfs, doi:10.3390/ijfs10040096_

Round 1

Reviewer 1 Report

Referee Report for “Optimal stopping methods for investment decisions: A literature review”

Manuscript ID: ijfs-1919311

Summary

I think the research is quite helpful. It presents a classified review of the literature on optimal stopping problems, the optimal stopping problems are important for decision-making and are less concerned by investors. The paper is well-structured, it provides a detailed description of the types of optimal stopping problems from different perspectives, such as the underlying process and the utility function, the analytic solutions corresponding to the problems are also included for better application. The paper also introduces some applications in the stock market, which show that optimal stopping methods are helpful. This article is logical and detailed on the whole, but I still have some concerns for you.

Major Concerns:

1.       The applications in Section 4 strategy performance are not abundant enough, as the formulas in the first three sections are complex, the performance of the strategies will more directly reflect the effectiveness of the optimal stopping strategy. More papers concerning the application of optimal stopping strategy should be included in Section 4.

2.       The primary references of the paper concern the closed form solutions to optimal stopping problems. However, due to the complexities in the real-world, analytically insoluble models are common in finance, deriving economic insights from a realistic numerical model of an economic system is preferable to deriving irrelevant answers from an unrealistic but solvable model. Therefore, I suggest adding some literature on numerical solutions in Section 4.

3.       For the conclusions of Section 5, future research on the optimal stopping problems can be expanded, for example, building the optimal stopping strategy based on behavioral finance.

4.       On page 7, line 198, the objective function is not used in Conrad(2018), it’s used in Xu(2021), a comment on Conrad(2018).

Minor Comments:

1.       On page 1, line 11, ‘optimal stop method’ is better to be replaced by ‘optimal stopping method’ to align with the title.

2.       On page 4, line 136, ‘’a’’ and ‘’b’’ are suggested to be in italics.

3.       Pay attention to capitalization, for example, ‘Investor’ on page 9, line 229

References:

(1)          Conrad, J. M. Real Options for Endangered Species. Ecol. Econ. 2018, 144, 59–64. https://doi.org/10.1016/j.ecolecon.2017.07.027.

(2)          Xu, J. A Comment on: “Real Options for Endangered Species” (Conrad, 2018). Ecol. Econ. 2021, 189, 107175. https://doi.org/10.1016/j.ecolecon.2021.107175.

Reviewer 2 Report

Referee Report for the manuscript, “Optimal stopping methods for investment decisions: A literature review” by Liu and Mu

The manuscript provides a detailed review of the current literatures on optimal stopping approaches and a practical guide for investment decisions in financial markets.

In this paper, the author(s) highlighted existing optimal stopping models and analyzed how these methods can be applied to financial applications. Practically, this paper provides new insights of practical relevance and decision support for both academics and practitioners. In general, the paper is well structured and well written in English. The most important thing is that as a literature review, this manuscript does very well in reference listing and the consistent style in references. The references list looks very clean and is above average standard.

Please see my major comments as follows.

Required Corrections

1. Discussions on the optimal stopping methods possibly used in other fields.

In this paper, the author(s) highlighted the optimal stopping models existed in the current literature and analyzed how these methods can be applied to financial decision. I know this paper aims to provide practical suggestions regarding the optimal stopping time for investment decision making. But I still suggest the author(s) to state how these methods are used in other research fields, e.g., medical science, engineering, computer science or other natural sciences. The author(s) may need one paragraph to talk about the applications in other fields (in the Introduction section). In this paragraph, the author(s) could also discuss why the financial market is a proper experiment for the use of optimal stopping methods, or does the market has its own unique advantages.

2. Table 1

I really enjoy reading Table 1 presented in Section 2.1, which provides a detailed description regarding the underlying process. I am wondering, if it is possible, that the author(s) could give a column of names for the underlying stochastic processes or models next to, let’s say, the first column “Label”. If author(s) worry about the limited space, try a horizontal landscape. For example, author(s) could rename the second process as “Geometric Brownian Motion” and so on.

3. Underlying processes

Please let me continue my comments on the underlying processes stated in Section 2.1. After Table 1, the author(s) provide a detailed discussion on the underlying processes. The author(s) classified all the processes properly and talked about the relative applications. The dedicated paragraphs should be properly enlarged, analyzed and developed regarding the comparison among these processes. The author(s) could discuss the advantages and disadvantages of each process in the revised version.

4. Links between equations

I have to say it is not a pure literature review. It contains more mathematical equations than other standard literature reviews. The author(s) spent huge efforts on the literature reading and discussions. My personally suggestion is that when trying to sell the paper (with several mathematical expressions) to the audience without strong mathematical background, please provide some links (descriptions or explanations) for equations to make them smooth. For example, on Page 8, before Equation (26), one could say that “Recall Equation (21), we have the expectation of the discounted payoff under the risk-neutral measure as: ...”.

Typos

As a literature review, it should present all mathematical expressions and notations consistently. I totally understand that the author(s) did a huge reading and collection of the current literature. Well done. But the author(s) should notice the notation consistency. For example, In Section 3.1, the author(s) use Xt to denote the asset price, which is a very commonly notation for stock price. However, in the next subsection 3.2, the author(s) changed the notation for stock prices from Xt to St, though it is common notation either. I know probably in reference [25], authors used Xt in their paper, but in other papers, authors consider different one. But when integrating everything in a literature review paper, please make them in a consistent way.

Another example is the notation for an indicator function: somewhere “ 1 ” is used to denote the indicator function, e.g., Equation (34); but somewhere “I” is used.

Pay more attention to the notations for “expectation”, “probability” and the brackets after expectation and probability.

If possible, please keep terminology consistent? In particular, “optimal stopping” v.s. “optimal stop”, e.g. Line 11 on Page 1.

I personally strongly suggest that the author(s) could double check the space between two sentences.

·         Line 14 on Page 1, I would like to say “After using real market data to describe how the market changes in a stochastic model, investors would predict the market movement in the future and make relative investment decisions based on the model.”

·         Line 17, I would like to replace “problem(s)” with “model(s)”

·         Line 18, I would like to say “The optimal stopping models suggests when investors should buy and sell, and the Merton’s model solves the problem of how much investors should buy and sell.”

·         Line 21, I would like to say “. . . and constraints”.

·         Line 23, I would like to split the sentence as “. . . as an example. Suppose that . . . ”.

·         Line 30, I would like to say “In an optimal stopping problem, we often obtain . . .

·         Need one sentence to link the citations [2, 3] and [4] to make a smooth transition.

·         Line 34, I would like to say “A martingale method for studying discrete-time optimal stopping problems is proposed by [4].”

·         Line 35, Concerning “The following research combines optimal stopping problems with dynamic programming problems.”, I only find one reference in the rest of this paragraph. How about to say “On the other hand, applying a dynamic programming framework to solve the optimal stopping problems could be more closed to the real world.”

·         Line 45, I would like to say “In this strand of literature, authors often pre-specify the shape of the boundary in a general form and then solve and verify it.”

·         The sentence from Line 53 is quite long. I would like to split it as “To address this issue, some subsequent studies naively added the time dimension into the model. However, it would increase the difficulty of solving a two-dimensional problem, because the free boundary problem needs to be solved by a partial differential equation. It is challenging to obtain an analytical solution for this problem.”

·         Line 61, I would like to split the sentence as “. . . the finite horizon. This proposed method also can be extended to other finite horizon problems.”

·         Line 52, I would like to say “In addition to adding the dimension of time, several studies investigate the impact of the maximum historical price on decision-making.”

·         Lines 67 and 69, I would like to replace “gave” with “provided” or “presented”

·         Line 75, “complex stochastic processes are considered

·         Line 77, I would like to replace “good” with “proper” or “decent”

·         Line 77, I would like to say, “In such flexible model, the parameters in each state are different, and the transformation time between different states is random.”

·         Line 80, I would like to say “is a type of regime-switching models”

·         Lines 82 and 82, I would like to use “in a regime-switching framework”

·         Line 85, I would like to say “From the practical perspective, this paper aims not only to provide a series of investment strategies based on optimal stopping, but also investigate the solutions of various stopping problems related to option pricing and derivatives hedging.”

·         Please try to avoid using “gave”. Maybe consider “presented” or “proposed”?

·         Line 90, I would like to split the sentence and say “However, this paper did not go through the relevant applications of investment decision making.”

·         Line 90, I would like to say “Inspired by the summary of the basic knowledge of optimal stopping problems in [21], some new development and application of optimal stopping problems are summarized as follows.”

·         Line 124, I would like to say “the choice of the utility function is essential in formulating the problem.”

·         Line 126, I would like to say “The optimal stopping problems can be classified into six categories as follows: the classical sequential analysis problem, disorder problem, optimal prediction problem, buy- low and sell -high problem, option pricing problem, and others.”

·         Line 135, I would like to say “the number of observations increases”

·         Please add a comma “,” after Equation (2), then say “where 1 is an indicator function, which is 1 if ..., otherwise 0. ”

·         I would like to use “bull and bear markets transitions” instead of “transformation”.

·         Line 138, I would like to say “In this problem, we generally define a new ...”

·         I personally suggest that the author(s) could use the Process ... with a capital “P”

·         Line 142, please delete “kind of”

·         Please add a comma “,” after Equation (4) , then say “where ...”. Please change Equation (5) as an inline equation.

·         Please keep consistent regarding the notation for “expectation”. Normally, we use E.

·         Line 150, I would like to say “From the perspective of investors, they aim to maximize their utility function instead of identifying the market transition time as soon as possible. The utility function is shown as:

·         Please note that we usually put a comma “,” before “respectively”, e.g., Line 153.

·         Line 160, I would like to say “The proposed optimal prediction model is applied not only for predicting the highest point, but also for predicting the resistance and support levels.”

·         Please add a comma “,” after Equation (8) , then say “where ...”.

·         Line 165, I would like to say “Assume that an investor is holding an asset, and a simple target is to find the time when the discounted asset payoff reaches the highest point to sell. Following [25], the utility function is shown as: ”

·         Line 167, I would like to say “Oppositely, if an investor intends to buy an asset, a simple target is to estimate the time when the discounted asset payoff reaches the lowest point to buy. Following [36], the utility function is shown as: ”

·         Line 169, I would like to say “In a market with trading costs, the investor ...”

·         Please keep Equation (12) consistent with other equations (use sup or inf) like (10) and (11).

·         Line 176, please illustrate the patterns of American options. And start a new sentence for Russian options.

·         The title of Table 2, “Different types of options”

·         Line 187, author(s) need a new section for “other problems”. And please use “Other problems” instead of “Some other problems”.

·         Line 202, please use “identify” or “decide” instead of “judge”.

·         More descriptions or explanations for Equations (26) and (27)?

·         Probably need a “, where” between Equations (30) and (31)

·         Line 238, “The optimal stopping time is solved as:”

·         Line 241, “As for the objective function given by Shiryaev”

·         Please add a comma “,” after Equation (46) , then say “where ...”.

·         Line 259, in a literature review, please specify the definitions of notations, instead of telling the readers to refer the literature.

·         In Equation (49), I think the range for τ is missing, and the same problem in Equation (52).

·         Line 261, I would like to say “Recall Equation (49), now we have: ...”

·         Line 269, “An important question in the asset pricing area is to find their resistance and support levels when investors conduct the technical analysis.”

·         Line 270, “The game between buyers (demand) and sellers (supply) often leads to an upward or downward trend in asset prices.”

·         Line 275, “construct effective trading strategies and gain extra earnings.”

·         Line 275, did the reference [35] use the optimal stopping methods? If yes, please discuss with more details; if not, please remove the sentence, it is not relevant to this paper.

·         Line 279, please add $$ for F. It is a notation.

·         Please rewrite Equation (58), I am a bit confused about Set G.

·         Line 306, “discounted asset payoff”

·         Line 337, “In the real financial market”

·         Line 337, I would like to say “the asset price would not always stay in a fixed upward or downward trend, but would appear alternately in the combined form of the bull market, bear market, and consolidation period.”

·         Equation (81), please make the equation consistent with other stochastic processes by using St instead of S(t)?

·         Line 356, I would like to say “Consider an investor who holds a share of stock with the price at time”

·         More descriptions or explanations for Equations (94), (95) and (96)?

·         Line 374, please give one sentence to differ the case 1 and case 2? Maybe “To solve the problem in a real world, in this case, we consider a suboptimal solution. ”

·         Line 385, please use “ηi, for i = 1, 2, 3, 4,”, instead of ηi(i = 1, 2, 3, 4).

·         Line 390, I personally suggest to add one sentence to motivate why we need Section 3.5.3.

·         Line 396, kup and kdown represent the number of rising and falling regimes?

·         Line 397, Nup and Ndown represent the total magnitudes of increase and decrease?

·         Line 401, I would like to say “The previous part summarizes the construction process of the optimal stopping strategy from the theoretical aspect. Practically, a strand of literature applies the optimal stopping method to the financial markets and evaluate how the optimal stopping strategies perform.”

·         Line 409, I would like to say “The strategy can provide an efficient stop-loss signal in several specific periods, such as the 2008 financial crisis and the European debt crisis. In some emerging markets, for example the Chinese stock market, by following the optimal stopping strategy, investors are able to stop their losses when the stock market crashed in the financial crisis.”.

·         Line 413, I would like to say “investors who employ the optimal stopping models only lost”

·         Line 415, Please replace the semicolon with “.”?

·         Line 427, Please use “Case 2” and “Section 3” with capital letters?

·         Line 441, I would like to say “the first is the selection of the underlying process,”

·         Line 442, I would like to replace “Secondly” with “Second”.

·         Line 447, I would like to say “It is more challenging to obtain the analytical solution given a relatively complex utility function.”

·         Line 463, I would like to start the paragraph as “However, several relevant research problems remain unexplored by the current literature. One potential problem with the existing optimal stopping strategies is that they cannot be used for multi-asset decision-making.”

·         Line 471, I would like to say “Another natural extension could be the choice of the objective function in investment decision making. ”

·         Line 483, I would like to say “With the substantial development in the optimization of numerical algorithms and the optimal stopping theory, researchers and practitioners can combine different objective functions with different stochastic processes. Thus, they can put forward new investment strategies based on the proposed optimal stopping methods.”

Conclusions

The subject is interesting, the article main underlying idea is classical; it is well built (successive parts built for writing a story), and is full of various elements. Finally, it gives reasonable insights for academics and practitioners.

It would be better that the author(s) could revise the paper based on my major comments and correct the minor typos listed above. I will be pleased to recommend the publication of this paper after a minor revision.

Reviewer 3 Report

The topic of the paper is interesting, but some suggestions are provided in order to improve the work:

Abstract

The methodology part is missing and also the innovative contribution of the work with respect to the existent literature on the topic

Introduction

This section seems a description of the methodology instead of a topic's analysis. I suggest to divide the two part: the first (introduction) could discusses the importance of the stopping method analysis or the role of the time in the investment decisions, and maybe the second part could contains the literature review and finally the third part the description of the methodology/ies and then the dicussions and conclusions. Moreover, in the initial analysis of the issue the role of the time in the investment decisions must be deply examined. Some useful references can be: Locurcio, M., Tajani, F., Morano, P., & Anelli, D. (2021). A multi-criteria decision analysis for the assessment of the real estate credit risks. In Appraisal and Valuation (pp. 327-337). Springer, Cham and Kou, G., Peng, Y., & Wang, G. (2014). Evaluation of clustering algorithms for financial risk analysis using MCDM methods. Information sciences, 275, 1-12.

Round 2

Reviewer 3 Report

The efforts made by the Authors are apprecciated